# The heartbeat of the city

**Rafael Prieto Curiel**[1]*, **Jorge Eduardo Patino**[2], **Juan Carlos Duque**[2], **Neave O'Clery**[1]

**1** Centre for Advanced Spatial Analysis, University College London, London, United Kingdom, **2** Research in Spatial Economics, Universidad EAFIT, Medellín, Antioquia, Colombia

* rafael.prieto.13@ucl.ac.uk

**Data Availability Statement:** The data is owned by the Mexico City Government which grants anyone access to all the data we used. Open Access data available at https://datos.cdmx.gob.mx/explore/dataset/carpetas-de-investigacion-pgj-de-la-ciudad-de-mexico The code which can be used to compute the heartbeat of some events with their

## Abstract

Human activity is organised around daily and weekly cycles, which should, in turn, dominate all types of social interactions, such as transactions, communications, gatherings and so on. Yet, despite their strategic importance for policing and security, cyclical weekly patterns in crime and road incidents have been unexplored at the city and neighbourhood level. Here we construct a novel method to capture the weekly trace, or "heartbeat" of events and use geotagged data capturing the time and location of more than 200,000 violent crimes and nearly one million crashes in Mexico City. On aggregate, our findings show that the heartbeats of crime and crashes follow a similar pattern. We observe valleys during the night and peaks in the evening, where the intensity during a peak is 7.5 times the intensity of valleys in terms of crime and 12.3 times in terms of road accidents. Although distinct types of events, crimes and crashes reach their respective intensity peak on Friday night and valley on Tuesday morning, the result of a hyper-synchronised society. Next, heartbeats are computed for city neighbourhood 'tiles', a division of space within the city based on the distance to Metro and other public transport stations. We find that heartbeats are spatially heterogeneous with some diffusion, so that nearby tiles have similar heartbeats. Tiles are then clustered based on the shape of their heartbeat, e.g., tiles within groups suffer peaks and valleys of crime or crashes at similar times during the week. The clusters found are similar to those based on economic activities. This enables us to anticipate temporal traces of crime and crashes based on local amenities.

## Introduction

The increased level of social interaction in cities is a key ingredient for the spread of ideas, technology, innovation and capital, but it can also be one of the main drivers of disease spread, crime and violence [1, 2]. In particular, security is a pressing issue for many cities around the globe. Despite efforts to reduce the frequency or mitigate the impacts of crime or crashes, there is a long way to go in terms of creating safe streets and neighbourhoods for people to walk and to live in. Understanding the conditions which increase the likelihood of a crime or a crash occurring helps us better understand and tackle such insecurity. If, for example, a street has many crashes at night, it could be due to reduced visibility and increased driving speed. If a square has many crimes in the afternoon, it could be related to the high flow of people

corresponding time can be found at: https://github.com/rafaelprietocuriel/heartbeat.git the authors confirm that others would be able to access these data in the same manner as the authors and that the authors did not have any special access privileges that others would not have.

**Funding:** This project received financial support from the PEAK Urban programme, funded by UKRI's Global Challenge Research Fund, grant no. ref.: PES/P011055/1

**Competing interests:** The authors have declared that no competing interests exist.

observed during lunch hour. But reduced visibility on the street, or increased driving speed or flow of people during lunch hour, is roughly the same from one day to the next one, and more specifically, from one week to the next one. Furthermore, the rush observed on a Monday morning is likely to be similar to the rush observed on a Tuesday or Wednesday, and less similar to a Saturday or a Sunday morning. Hence, repeated patterns and conditions are most accurately observed on a weekly basis.

In this paper we explore the occurrence of two predominantly urban phenomena, crime and crashes, through the lens of weekly cycles. We construct a new mathematical function for this analysis, termed "heartbeat", at a city and the local level, and apply it to study these phenomena in Mexico City. In this section we review the literature concerning urban insecurity, and the spatial and temporal analysis of crimes and crashes.

## Urban insecurity

Crime is one of the most pressing problems around the world. Globally, more than 1,000 people fall victim to an intentional homicide each day, nearly three times the number killed in armed conflict and terrorism combined [3]. In Latin America, crime and fear of crime have been identified as the most important issue to citizens, over unemployment or access to health [4]. The direct cost of crime in Latin American economies is at least 3% of their GDP, equal to their annual spending on infrastructure [5]. Whilst crime is suffered in rural areas as well, crime is mostly an urban problem [6]. In Mexico City, for instance, there were more than 60 crimes for every 100 inhabitants in 2018 (with 38 crimes for every 100 inhabitants at a national level) [7]. Crime is similarly high in other cities, thus it is not surprising that in Mexico, 73% of the urban population fears crime [7, 8]. Furthermore, not only is crime mostly urban but also, larger cities suffer disproportionately more crime [1, 9–12].

Whilst crime and violence are one of the main urban issues, road accidents kill more than 1.35 million people around the world each year while 50 million people suffer non-fatal road injuries [13]. Cars kill nearly three times more people than all types of crime and violence combined. Most of the casualties related to crashes worldwide are economically productive young adults and so crashes mean lost productivity [14]. Reducing road deaths is the focus on Target 3.6 of the Sustainable Development Goals, which aims to halve the number of global deaths and injuries from road traffic accidents by 2020. This target will not be met. Considering the medical and emergency costs and property damage, the economic burden of crashes is of a similar magnitude to crime [15, 16]. When the impact on the quality of life from crashes is considered, the harm from motor vehicle crashes becomes even larger [17]. Crashes appear to be on the rise in urban areas relative to rural settings. In the US, for instance, 54% of the vehicle crash deaths in 2018 were urban, up from less than 40% in 2000 [18]. Overall, non-fatal accidents are more frequent in urban areas, and the number of accidents grows superlinearly with city size [19]. As with crime, road insecurity is a major and costly urban concern.

## Crime and crashes as spatial and temporal events

Crime and crashes place a huge burden on society, despite the efforts of authorities and citizens to prevent them and to mitigate their impact. Challenges to prevention and mitigation include data collection and interpretation, but also, the development of suitable modelling techniques which can be used to predict and prevent future incidents and design security policies. Crime itself is complex since there are many nonlinear feedback loops, with self-organising agents, which give rise to system-wide unexpected behaviours which are difficult to understand and control [20] including issues with rehabilitation and recidivism [21], punishment [22] retaliation, [23] and others [20, 24]. Similarly, although drivers try to avoid

collisions, there are some spatial and temporal repeating patterns in terms of car crashes and also with pedestrians and cyclists. Whilst traffic is one of the most relevant measures to explain density of accidents [25] with more traffic, congestion happens, which in turn, reduces speed and the lethality of accidents as well [26]. The accident-prone locations are mostly located in districts with higher speed and traffic volume [27].

Attempts to discover relationships between crime and social factors using statistics may be traced back to Adolphe Quetelet and André-Michel Guerry, who coined the term 'Social Physics' in the 19th century [24, 28]. Quetelet and Guerry studied conviction rates, using tools from astronomy to measure the "true level of criminality" [29]. Simeón-Denis Poisson investigated wrongful convictions to develop what we now know as the Poisson distribution. Since then, many patterns have been recognised in terms of crime. We know, for instance, that crime is highly concentrated among many of the units in which it can be observed [30], including victims [31], perpetrators [32], their families [33], in space [34–36], and across cities [37]. The fact that crime is concentrated results in hotspots of crime, which has many implications in terms of security [38]. Road accidents are also concentrated in space, and form hotspots [25, 39], usually nearby highways and major junctions, that are not just the result of randomness [40]. The accident hotspots also give relevant patterns in terms of where they happen, for example, that most pedestrian injuries are situated on major roads [41].

The temporal dimension is as relevant as the spatial dimension. In terms of crime, yearly trends [42], seasonality [43, 44], some periodicity [2, 45], weekly cycles [46] and the relationship with economic cycles [47] have been studied. With respect to crashes, it has been noted that their frequency and severity exhibits daily cycles [48], and that the peak of traffic accidents coincides with the peak traffic congestion on arterial roads [27], among other temporal patterns. Both time and space are relevant dimensions to analyse crime and crashes, which interact. A hotspot map is not a static description of the patterns of crime or crashes in a city, but it reflects the interval of time considered. Most likely, the hot areas shift if different periods of time are considered. For instance, the map of Friday evening events might reflect leisure activities and could be distinct to the Monday morning map, which might reflect work or school routines. Similarly, if crime rates drop in a city, it does not mean every neighbourhood experiences fewer crimes (some could even experience more), but that temporal trend is the result of some spatial aggregation. For the analysis of events which exhibit temporal and spatial dynamics, such as crime or crashes, specialised tools based on the spatio-temporal aspects of such events are needed.

## Weekly units of time dominate social patterns

Perhaps the most relevant temporal unit to analyse social events is weeks. On a Monday morning, students go to school and workers endure the rush hour. Then we eat lunch at roughly the same time, and so restaurants get busy. Later, cafes and bars become busy and commuters go back home, forming an opposite traffic wave to the morning. On Tuesday and Wednesday, the routine is similar. Although weekdays are not exactly the same and routines are not exactly repeated, Mondays are not substantially different from Thursdays. Fridays, however, mark the beginning of the weekend and the beat of the city is different. Social life is increased and cities usually remain awake for longer hours. During the weekend, museums, parks and other amenities get busier. Financial districts and areas around big office buildings become a ghost town on Saturdays and Sundays. In this way, cities synchronise activities and routines on a weekly basis. It is only natural, therefore, to expect weekly cycles with respect to patterns formed by most social events, such as the number of customers in a shop or the number of people streaming a video. Weekly cycles, related to patterns of work, education and leisure, should also be

expected when it comes to incidents such as crimes or crashes. For example, the likelihood of a crash is increased by cyclic patterns in commuting activity, the density of pedestrians and cyclists on the street, alcohol consumption or even darkness.

Identifying whether crime has weekly cycles is a key element in determining the demand for resources and for policing. At the city level, it enables advance planning of the expected demand of security resources. At a local level, knowing that certain neighbourhoods have different crime intensities throughout the week enables an allocation of resources closer to crimes, and a reduction in response time to 911 calls. In the case of crashes, detecting cycles also aids the allocation of emergency resources. At a local level, it further provides insights into the plausible causes of accidents and ideas on how to prevent them. If, for example, a neighbourhood has an increased number of accidents during the mornings from Monday to Friday, then the chances are that they occur in a school or office district, and that rush hour contributes to that higher intensity. However, if the neighbourhood has more accidents on a Friday or Saturday night, then the chances are that they are related to leisure and alcohol consumption. Yet, weekly cycles are frequently ignored since time is usually analysed in a linear manner (based on mean rates, trends and departures, so that time indicates the sequence of events) but not considering whether events, although perhaps many days apart, tend to happen on a Friday night, for example.

Furthermore, although crime and road accidents are substantially distinct social events with no apparent or direct link, they result in similar burdens in terms of city resources. Serious accidents require ambulances and paramedics, but also the attention of the police to cordon off an area and divert traffic. Crimes also require the attention of the police, but often crime (particularly violent crimes) also need the aid of ambulances and paramedics. Therefore, a city might suffer a slower crime response if there are too many crashes simultaneously, and might suffer delays in the response to crashes if the city has too many crimes. Hence, this unexplored temporal correlation is of utmost policy importance.

Data from Mexico City gives the location and time of crashes detected by the Emergency Attention Centre and crime reported to the police. City tiles are constructed using the location of public transport stations in a novel way using a distance-between-stations parameter. This method divides the city in a more organic way while allowing us to test for the modifiable areal unit problem. Weekly cycles are detected in terms of the heartbeats from both types of events and the observed peaks and valleys are used for classifying the tiles of a city. We then use economic activities in the city for classifying tiles and observe a high level of similarity between the tile groupings (one based on the heartbeat of crime or crashes, and the other based on economic activities).

## Detecting spatio-temporal patterns in crime and crashes

### Hotspots and cycles of crime

According to *Routine Activity Theory*, crime happens when a motivated offender, a suitable target, and the right conditions (such as the absence of guardians) converge in space and time [49], and in settings that make committing the crime easy, safe and profitable [50]. Routine Activity Theory suggests that understanding crime patterns relies on understanding why offenders, targets and conditions converge in time and space and argues that prevention can be achieved by altering any of these three elements [49]. In terms of opportunities, some parts of a city attract a large number of people and facilitate crime via large concentrations of people and targets at particular times and places. For instance, a prime location might be a public transport station or a busy stadium (named "crime generators" by Patricia and Paul Brantingham). Some parts of the city attract motivated offenders, e.g., nightlife or prostitution areas,

which draw intending criminals (named "crime attractors") [50]. These settings, which generate profitable crimes, keep repeating: the busy square today will be just as busy tomorrow; the dark alley will be just as dark next week; the distracted customers leaving the bar will be just as distracted next month. In space, crime is mostly concentrated in non-random locations [36] which are somehow stable [51]. Keeping everything else equal, changes in routine activities can alter the likelihood of convergence of offenders and targets in space and time, altering crime outcomes [49]. Everyday life in cities, characterised by routines, constrains the convergence of opportunities, and helps to explain why crime tends to occur in recurrent spaces and times.

The spatial concentration of crime, enhanced perhaps by the presence of attractors and generators, produces areas of the city with a high ("hot") and low ("cold") intensity of crime. We can present these hotspots visually on a map [52, 53]. The presence of attractors and generators of crime also produce periods of time with higher or lower intensity. If a public transport station generates crime because it is busy, it generates crime during rush hour, a busy stadium generates crime when there is an event, and a district with many bars attracts crime when bars are operating. Thus, the temporal concentration of crime is embedded in the spatial concentration of crime, and hotspots on a map have a counterpart as peaks or valleys on a time series.

Analysing temporal patterns of crime has played an important role in crime analysis for decades. For example, yearly crime rates between 1898 and 1926 in some US states were analysed to detect cycles in crime [54]. More recently, time series have been used to detect seasonal cycles in crime [44] or to trace serial crimes [55]. Some advanced mathematical tools have been applied to the analysis of crime cycles, for example, Fourier analysis was used in crime data in a city in South Africa to show that violent crime peaks every 75 and 150 days [45]. Weekly cycles of crime have been explored before, using "circular statistics" [46], with temporal variations consistent with patterns implied by routine activity theory [56].

Spatio-temporal patterns of crime have also been an object of study. For example, for optimal policing [57] or to study demographic covariates and their impact on violence interruption [58]. It has been noted that different types of crime exhibit seasonal patterns, which vary across space [59]. Data from Chicago showed that high schools appear to attract robberies around school hours, but that street robbers seem to perpetrate in transit hubs most of the time irrespective of how many potential victims are around [60]. Data from Campinas, Brazil was used to show that crime exhibits spatio-temporal patterns, which vary according to different types of crime [61] and data from Florianópolis, also in Brazil, was used to show that street robberies' hotspots change across space depending on the time of the day [62]. Even a plot triplet made of a hotspot map, a yearly trend -with weekly data- and a daily trend -with hourly data- was suggested as a tool to analyse crime patterns [42].

One of the key drivers of why high schools attract robberies in Chicago, or why street robberies shift depending on the time of the day in Florianópolis, is due to daily commutes. In Vancouver, for example, some suburbs double their nighttime population whilst others lose half due to daily routines [63]. In a city in Eastern Canada, it was observed that 7% of census tracts concentrate one-half of work visitors and 9% concentrate one-half of the shop visitors. Hence, it is not surprising that 23% of census tracts concentrate half of the property crimes from that city [64]. Commuters can have a major impact on crime [65] and since commuters have very clear daily and weekly cycles, spatio-temporal analysis of crime should also view space and time based on daily and weekly cycles. Notice it does not mean using daily or weekly data, as it is frequently used in crime analysis, but it means to analyse time as a cyclic variable, that is, to take into account that there are Mondays and Fridays, there are weekdays and weekends.

## Hotspots and cycles of crashes

Crashes have three causal factors: mechanical failures, environmental factors and human error [66]. Mechanical failures include, for example, a sudden brake failure, bald tires, or other vehicle defects. Spatial or environmental factors include a faulty traffic light or street network whose geometric features increase the probability of accidents [67]. Yet, most accidents happen because of human error, mainly carelessness of drivers or pedestrians [68] arising from fatigue and sleepiness [69, 70], distraction or alcohol [71] or, most likely of all, speed [26]. At lower speeds, crashes become much less severe (a 10% reduction in the mean speed of traffic results in a reduction of more than 30% of the number of fatalities, and more than 20% in the number of seriously injured victims [26]). In many cases, a combination of these factors all come into play [72].

Traffic is a key factor in the density of accidents in a place [25]. On streets with large amounts of traffic (both higher speed and traffic volume), more accidents are expected [27, 73]. Therefore, since traffic is concentrated in certain areas, and since environmental factors and some human errors occur more frequently in some areas than others, crashes are concentrated and form non-random hotspot patterns [40]. These include, for example, areas with a high density of pedestrians in the dark, or where there are "careless weekend drivers" [39]. Alcohol consumption by pedestrians increases their risk of collision with a vehicle and so pedestrian injury hotspot are often in areas with a high density of alcohol-serving establishments [41]. Crashes form patterns in space.

The temporal distribution of crashes is also important. It has been noted that the peak of traffic accidents occurs when there is congestion peak hours on arterial roads [27]. For example, in Shiraz, Iran, between 12:00 and 14:00 there is a peak in the number of accidents, and between 4:00 and 6:00 there is the valley of accidents [48]. Spider (or polar) plots have been used to report daily and weekly cycles of crashes [27, 48]. Furthermore, as in the case of crimes, the spatio-temporal patterns of crashes are an emergent property. If, for instance, alcohol consumption by pedestrians increases their risk of collision, we will likely observe hotspots both in neighbourhoods with a high density of alcohol-serving establishments and during the periods of the week when alcohol consumption increases [74]. Traffic accidents exhibit spatial and temporal cyclical patterns: human routines cause large traffic flows in certain places, at certain hours and days [75] and routines concentrate, at specific locations and periods of time, activities which increase our exposure to collision risk.

Therefore, despite the complexity of crime and of crashes, routine activities and daily commutes lead to the convergence of criminals and victims on the right conditions for crime to occur on a cyclic manner (days and weeks), and create traffic and increase the frequency of risky activities for a crash to occur, also on a cyclic manner. Therefore, we expect spatial patterns to occur in a cyclic way when crime or crashes are observed, but these cycles have been frequently ignored. In previous studies of crime and crashes, time is usually recorded in daily cycles (which treat all days equally without distinguishing weekdays and weekends) or weekly cycles (which tend to overlook daily and within-day patterns). Here we analyse a 168-hour cycle encompassing the whole week, which enables us to uncover complex daily patterns and beats.

## Spatial analysis

The study of space and geography is a key part of the social science disciplines, from the analysis of the location of human activities to the analysis of relationships between social phenomena and the physical environment [76]. Spatial analysis is the analysis of data in which the location of observations and the distance between them is taken into consideration [77]. For

instance, studies focused on analysing the spatial distribution of crime have shown that certain land uses and population characteristics are related to crime hotspots [28].

A proper treatment of space requires recognition of the importance of the two-dimensional nature of spatial interactions (otherwise known as spatial auto-correlation) and the implications for statistical analysis [78]. A core presumption for researchers investigating any spatial phenomena is that everything is related to everything else, but physically proximate things are more related than distant things (Tobler's First Law of Geography, [79]). Therefore, although recording all factors present with respect to a single crash might be an impossible task since conditions might change rapidly [72], if we observe a cluster of crashes near alcohol-serving establishments, driving under the influence might be a common factor for all crashes. Similarly, many crimes which happen in close proximity could have a common environmental factor which converges criminals, victims and opportunities together.

It is important to bear in mind that our results depend on the spatial unit of analysis used. As such, the results may change if we modify either the size or shape of the spatial units. The sensitivity of the result to the choice of geographical units is known as the Modifiable Areal Unit Problem (MAUP), first introduced in the literature in the 1970s [80, 81]. Studies on the effects of MAUP have shown that effects vary according to the level of aggregation, i.e., the greater the level of aggregation, the greater the effects [82, 83]. The effects also depend on the spatial pattern of the variable under study, i.e., random patterns are less sensitive to MAUP than spatially autocorrelated patterns [84, 85]. Given that the MAUP is a common unavoidable problem when using aggregated data, the suggested practice is to design spatial units related to the phenomena under study. This ensures that the analysis is better connected to policy design, and minimises the aggregation bias [86–89]. In this sense, we designed our units of observation based on the flow of people (that is, the public transport stations) which satisfy both requirements since they capture population mobility patterns which are key components in both crime and traffic accidents [90, 91] and allow interventions at a local level.

## Constructing the heartbeats of crime and road accidents

### Data

We aim to investigate weekly cycles of crime and of crashes. We use data from Mexico City. In order to do this, we use open-access data from January 2016 to March 2020. We restrict the sample to pre-pandemic time because we expect significant changes in patterns due to lockdowns.

Data on crimes and crashes is typically incomplete, for many distinct reasons. In Mexico City, only 6% of crimes are reported [92] which induces an unavoidable bias in terms of the victims, the types of crime, whether the criminal was arrested and many more types of bias. However, whether a crime happened on a Monday or a Friday and whether it happened at 9:00 or 15:00 should not alter significantly the probability that the victim reports that crime to the police. It is likely that some temporal bias is induced by considering only reported crime. Without many other sources of data to analyse, measure or correct this bias, we assume that the reported crimes reflect the general level of criminal activity in the city at any specific time of the week. In the case of some types of crime, particularly if the victim is not present (for instance, a parked car being stolen) the precise moment at which the crime occurred might not be clear. This uncertainty could be significant, if, for instance, the person notices their car is missing hours after they parked it (and often the time of the crime is registered as an estimate, when the car was last seen or when the person noticed the car was stolen). Therefore, to reduce this potential source of bias, only robberies which were reported to the police and in which the victim is present are included in our dataset.

Road accidents data is obtained via the Emergency Attention Centre from Mexico City, which includes 911 calls and crashes observed by CCTV cameras or directly by a police officer or an ambulance around the city. Non-serious accidents might not be attended by the emergency attention centre, but the number of under-reported accidents, in the case of serious and fatal accidents, is reduced as they require some emergency attention, such as an ambulance [93] and therefore, most of them should be included in the dataset. All accidents included are confirmed either by an officer or an ambulance in the location or by CCTV and the time when the report was started is considered as the time of the crash, which has a delay of few minutes.

We focus on the location and time of crimes and crashes. Roughly 200,000 crimes and 940,000 crashes are included, with time and xy coordinates (see the S1 File 1.1 for details of the data processing). In total, 1,551 days between 1st January 2016 and 31st March 2020 are observed, which gives us a daily rate of 143.6 crimes and 605.6 crashes.

For each event, whether a crime or a crash, the moment at which it occurred is stored as a decimal between 0 and 7, where 0 corresponds to Mondays at 0:00, 1.75 corresponds to Tuesdays at 18:00 and 5.9 corresponds to Saturdays at 21:36, for example. Notice that 0 and 7 mark the exact same moment of the week.

## From discrete events to a continuous heartbeat

Events such as crime or crashes are a point process (a discrete event) in time and in space, but the risk is not a point (in time or in space). We estimate the background rate (or intensity) by smoothing the data, which is the distribution of a crime or a crash happening at a moment through a week. We apply a kernel density estimate for smoothing the data, which is a technique often used for modelling crime [58, 94] and for identifying road accident hotspots [27, 73, 95, 96]. In the case of crime, spatial and temporal kernels are often used to estimate the background rate [58] or the propensity that a specific location has to crime given an inhomogeneous point process, or as a self-exciting process [94], whilst in the case of crashes, similar tools have been used to estimate the background intensity of crashes [27, 68, 97]. Here, instead of a spatial technique, we apply an additive Gaussian kernel for the time of the week in which the event occurs for smoothing events in their time of the week.

Formally, if $t_i$ is the time of the week in which the $i$-th event happened, with $t_i \in [0, 7)$, then for time $t$, also in $[0, 7)$ which is the domain of $H$, the heartbeat $H(t)$ is given by

$$H(t) = \sum_i \exp\left(-\frac{(t - s_i)^2}{2w^2}\right), \tag{1}$$

where $s_i = argmin(|t - t_i - 7|, |t - t_i|, |t - t_i + 7|)$ gives the smooth data its cyclic behaviour; $w$ is the bandwidth of the smoothing process and the sum is over all the events. Notice that for large values of the bandwidth $w$, the intensity $H(t)$ becomes flat throughout distinct times of the week, whereas for smaller values, the function peaks around events $t_i$ and does not group events which happen at similar time.

The bandwidth chosen is large enough such that the propensity of crimes to be reported at even hours or 15-minute breaks is ignored, but small enough such that distinct moments of the day are identified (see the S1 File 1.2 for more details). The value of the bandwidth $w$ is such that if an event happens at time $t_i$, then our estimation for the background intensity gives a probability of one at time $t_i$; two hours later ($t_i + 2/24$), the estimated probability conditional on event $i$ decrease to 0.2 and after four hours, the chances are nearly zero, but 30 minutes before or after the event, the chances remain above 0.96 (see the top two panels of Fig 1 for a graphic description of the smoothing process of a small simulated dataset).

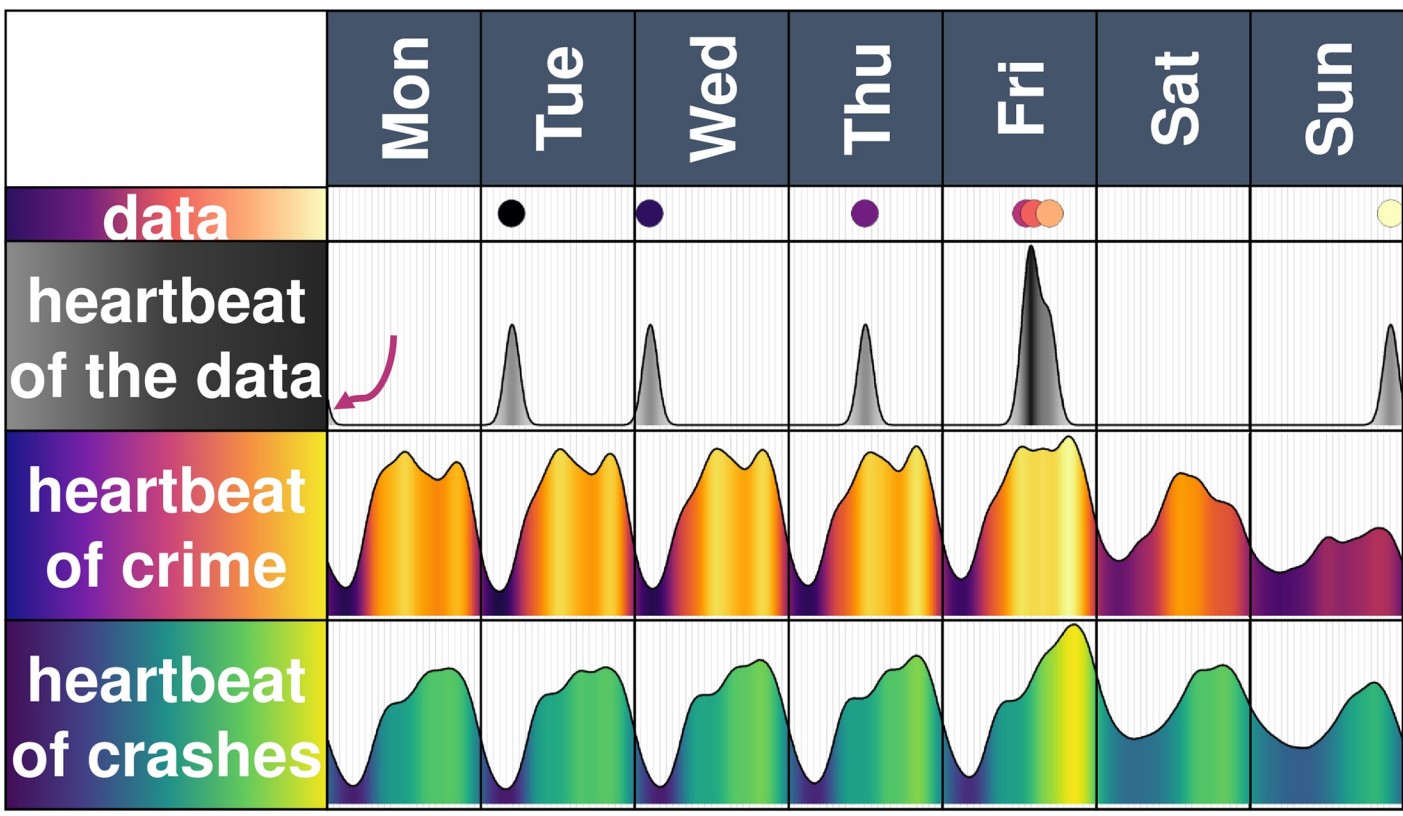

**Fig 1. Smoothing cyclic data via a kernel density.** The top two panels has 7 simulated data points and its heartbeat. Each point gives a smooth distribution which are then added (thus, the three points on Friday give a higher intensity as they are combined). Notice that the point on Sunday night also affects the heartbeat on early Monday (the arrow) as we are considering cyclic data. The third panel is the heartbeat of crime considering crimes in Mexico City. The last panel is the heartbeat of crashes from Mexico City.

Crimes have a daily and a weekly beat. The smoothing procedure described, applied to the 200,000 crimes, shows a pattern, which is nearly the same from Monday to Friday, which peaks at around mid-day and at 20:00 and then Saturday has its own pattern, which begins with a "warm" night, peaks at around 16:00 and ends with a warm night as well. Sundays remain "cold" throughout the whole day, although there is a peak at around 20:00 as well (see the middle panel of Fig 1, which is the function $H_C(t)$ applied to the time of the 200,000 crimes). The observed shape of the function $H_C(t)$ resembles an electrocardiogram and so we refer to the function $H_C(t)$ as the *heartbeat of crime*.

Road accidents have a daily and a weekly beat as well, with a low intensity of crashes from 0:00 to 8:00, a medium intensity throughout the day and a peak at around 20:00, which is more visible on Friday. Whilst Saturday and Sunday do not have a peak as high as Friday, their intensity of crashes is not as low during the nights as the rest of the week. The function $H_R(t)$ applied to the 940,000 crashes gives us the *heartbeat of crashes* in the city (see the bottom panel of Fig 1).

## Producing city tiles

Although the heartbeats of crime and crashes at city level help us inform an overall pattern of both events, we wish to construct them at a more local level, detect if they are the result of a mere thinning process from the data of the whole city or if the heartbeats at a local level are

substantially -statistically- different and if they inform aspects of the local attributes. It is possible to use a regular grid superimposed on a city to analyse the spatial pattern of crime and crash events (known as the quadrant count method [28]). Yet, grids often divide neighbourhoods into two or even four parts, and merge parts of a city in a less-than-natural way. This can lead to the presence of spurious spatial autocorrelation in nearby cells [28]. Using existing legal or political definitions for neighbourhoods or districts often yields a more natural partition of a city. Yet, the dividing line between two distinct neighbourhoods is often a main avenue, and the edge between three of more neighbourhoods is often a main city node (such as a Metro station) which functions as an attractor node of crime and crashes [40]. This results in crucial gravity centres of the city being divided into distinct observation units. Instead of using pre-existing partitions of the city, or an artificial grid, we use the layout of the public transport system to divide the city into 'tiles' and consider them our observation units. Metro and other public transport stations offer a natural way to delineate distinct parts of a city. For example, a Metro station in the financial district of a city attracts a specific population group at precise moments of the week, and has a rush hour (inside and around the station) at very different days and hours than a station in a residential neighbourhood or a station near a University.

Our procedure for constructing city tiles based on the public transport system is as follows. Firstly, all (major) public transport stations are included (e.g., Metro, BRT, trams and buses) which gives us a little more than 5,000 points in the city. Metro and BRT stations (8.7% of the public transport stations in Mexico City) are the leading public transport providers but some parts of the city are not well covered by these two systems. Therefore, including trams and buses gives us better coverage, particularly in areas less-served by the Metro. However, some stations, particularly from different systems or lines, might be located in close proximity (for example, 21% of the stations have another station within less than 50 metres), so we filter these out as they would give us tiles which are too small.

In order to select which stations to keep, we set a threshold, $\tau > 0$, which is the *maximum distance between stations*. If any two (or more) stations are at a distance smaller than $\tau$, then only one of them is kept. If the two stations are from different systems, then Metro, BRT, tram are picked, in that order, over buses. If the two stations are from the same system, then one is randomly picked. The method enables us to test different distance thresholds, and investigate the impact that the choice of $\tau$ has on the tiling and on the corresponding heartbeats. See the S1 File 1.4 for an analysis of the impact of the distance parameter $\tau$. Also, some of the spatial analysis covered here are analysed for different values of $\tau$ in the S1 File.

Once the final set of stations has been generated, we compute the voronoi partition of the city associated with each station. This corresponds to the parts of the city closest to that station than any other [98] and crimes and crashes that occurred inside the tile (or the nearest crimes and crashes to a station) are used to construct the corresponding heartbeat of the tile (Fig 2). Constructing the voronoi tessellation is a technique which has been used before for the spatial analysis of crime [99], crashes [100] and emergency management [101] since the technique easily allows detecting which ambulance is nearest to an emergency, for instance. The centre of each tile (the station) is used in order to label the tiles, so that the city has tiles called Zócalo, or Polanco, named after the Metro Zócalo and Metro Polanco, for example. The voronoi partition of the city gives a non-overlapping tiling of the city, so that all crimes and crashes are assigned to only one tile. Tiles have a different size, so crimes and crashes per unit area are used to compare the intensity of both. Although most of the tiles have roughly the same area (particularly near the city centre), the tiles near the edge of the city might not have all boundaries defined (thus, have considerably large areas and not be part of Mexico City), so the area of the tiles is trimmed for the 30% largest tiles.

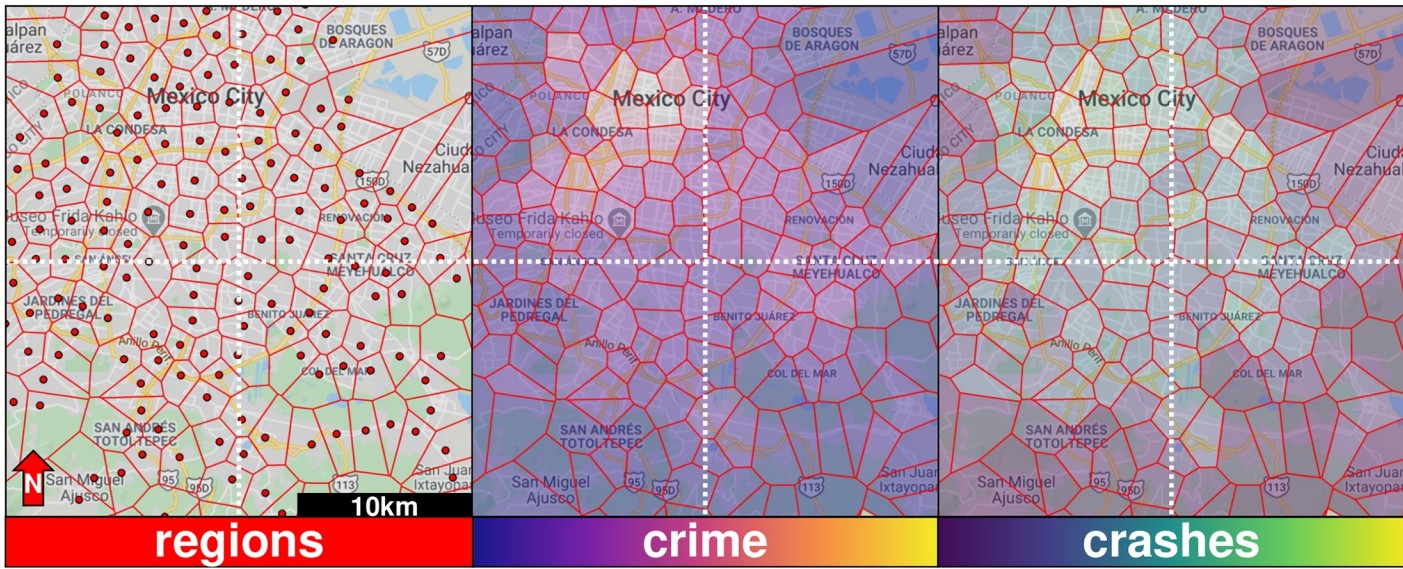

**Fig 2. Tiles of the city.** The left panel shows the central part of Mexico City divided into 213 tiles using a distance threshold of $\tau = 1,500$ metres. The red dots correspond to the public transport stations, which are the centre of each voronoi tile, meaning that for each polygon, the nearest public transport station is the red dot inside. The middle panel is the crime intensity per unit area, where lighter colours represent a higher intensity. The right panel is the crashes intensity per unit area, where darker parts have less crashes. Notice that the more peripheral parts of the city have less intensity of crime and crashes, but there are some areas (particularly East and South Mexico City) with a high intensity of crashes, but not a high crime intensity.

Our results depend on the spatial unit of analysis used in this study (i.e., the tiles based on the public transport system). As such, the results may change if we modify either the size or shape of the spatial units. In this sense, our tiles based on the public transport system capture population mobility patterns which are key components in both crime and traffic accidents [90, 91]. It is possible to easily identify tiles based on the corresponding Metro or BRT station and so our results can be used for local interventions in those areas and so they are more natural units of observation since they are connected with crime and crashes through their local dynamics. In the S1 File, we have included an analysis of the results which we show here for other values of the distance parameter $\tau$, which gives us more (smaller) units of observation.

## Constructing tile heartbeats

The heartbeats of crime and crashes are computed for each tile, considering the time of its events. We use the same partition for crimes and for crashes, so for example, with a value of $\tau = 1,500$ metres, it gives us 213 tiles, so 213 crime heartbeats and 213 crashes heartbeats. The height of the heartbeat depends on the intensity of crime (or crashes) in that tile and the temporal concentration. Investigating the intensity of distinct tiles is a hot spot analysis, but here, we are interested in the shape of the heartbeats (that is, when in the week it has a peak, a plateau or a valley) rather than the intensity.

The Pearson correlation between the heartbeats of separate tiles is a natural way to compute the synchrony of their events (and ignores the mean intensity). A high correlation (close to 1) means synchronisation of when events happen on both tiles. A low correlation (close to -1) means that peaks and valleys happen in opposite times, so that one tile gets hot whilst the other gets cold. A correlation close to 0 means that the peaks or valleys of the tiles are not simultaneous.

### Are the heartbeats distinct?

Could it be that the heartbeat of crimes (or crashes) from one tile is the same as the heartbeat from another tile (or even the whole city), but there are simply a distinct number of crimes or crashes in the second tile? Are the heartbeats, statistically speaking, distinct? We construct a test considering a pair of tiles with a high correlation on their crime heartbeat (0.9606) and produce 500 simulated heartbeats by sampling the data from crime in the city (meaning that we randomise space but keep the time signature of events). Under a null hypothesis, we could eventually reproduce the heartbeat that gives such a high correlation by just sampling data from the city, but we see that randomness does not produce such heartbeats. Thus, we reject the null hypothesis that the correlations could be the result of randomness (see the S1 File 1.3). We apply the same test to the crashes heartbeats and reject the null hypothesis as well. The crime heartbeats and the crashes heartbeats are statistically different and are a result of an underlying temporal pattern of the overall crime and crashes, but also, the result of a spatial process by which the heartbeats of one tile can be statistically different to other tiles and are not the result of random crimes. Furthermore, the test also confirms that we can use the correlation between two heartbeats to compare their crime and crashes temporal patterns.

## Results

### Weekly patterns of crime and crashes

The city has a heartbeat in terms of crime. From Monday to Thursday patterns are similar: mornings and evenings exhibit an increasing intensity, reaching their peak by 12:00 and 20:00 (Fig 3). Notice, however, that Mondays are slightly different in the mornings, as the crime intensity is 80% of the weekly peak at 8:00, but is just above 50% of the weekly peak on other weekdays. Fridays have a high intensity from 12:00 until late, when the city "cools down", but not as cold as the rest of the nights. Saturdays are different from the rest of the week, as it starts slightly hotter and reaches its peak at around 16:00, although a smaller peak than the rest of the week and a hotter night than weekdays. Finally, in terms of crime, Sunday is cold throughout the day and reaches a peak at around 12:00 and a second peak at 20:00.

In terms of crashes, the heartbeat exhibits a very similar pattern from Monday to Thursday. Fridays have a distinct pattern, as the number of crashes in the city increases significantly (compared to other weekdays) after 15:00 and remains at a peak until before midnight. Saturday is hot in the early hours, and is the hottest night of the week, although the rest of the day is not as hot as Fridays. Sundays also start hot, particularly at around 4:00, but then the day turns cold. Notice that the peak on Friday from 17:00 to 23:00 is surprisingly high—no other time of the week reaches such a high intensity of road accidents.

Daily and weekly beats are both relevant in terms of crime and crashes. Just by looking at the temporal distribution of the 168 hours of the week, a distinct pattern is noticeable. In terms of crime, Saturdays, although they overall have less crime, do not mimic the temporal crime patterns of a Friday or Tuesday. Saturday has its own peaks at distinct hours. Mondays also have a distinct pattern compared to other days of the week. In terms of crashes, there are some similarities from Monday to Thursday, but the other days exhibit distinct patterns. Overall, crime has a high intensity from 10:00 to 22:00 but crashes have a high intensity from 14:00 to 23:00. This only happens at a city level, but the same does not hold at a more local level. At a tile, the exposure changes in space and time, likely related to land use patterns.

Although crime and crashes are completely different social events which are the result of distinct social problems, at a city level they both reach their hottest and coldest point simultaneously (Fig 4). On Fridays at around 20:00, the city has the highest intensity of both crime

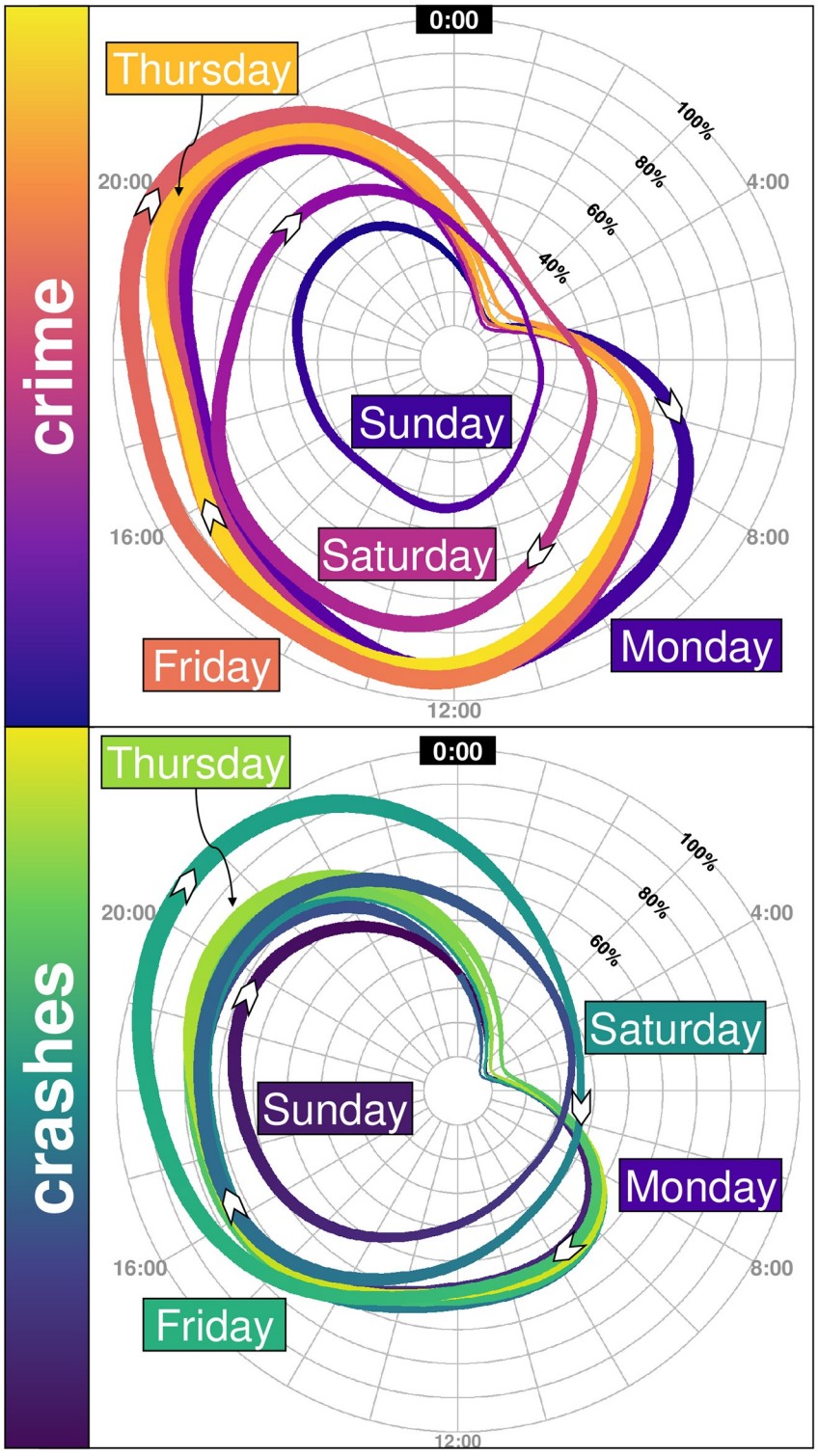

**Fig 3. Cyclic patterns of crime and crashes.** The top panel shows on a polar (or spider) plot (also invented by André-Michel Guerry nearly 200 years ago to display cyclic data) the varying intensity of crime in Mexico City, compared to its peak, meaning that the maximum is observed with a 100%, on Friday at around 20:00. The cycles show that Monday is slightly more intense during the mornings, but then the rest of the day follows the same patterns with a high intensity at around 12:00 and 20:00. Then Friday has a high intensity throughout the day and is more intense in the night than the rest of the days as well. The weekends are substantially different, their corresponding peaks are reached

at different hours (at around 16:00 on Saturday, and with roughly the same low intensity on Sunday from 12:00 to 23:00). The second panel shows the varying intensity of crashes, with a similar pattern from Monday to Friday morning (with nearly no crashes at around 4:00 and a similar peak from 16:00 to 22:00. Friday has a higher intensity and thus the city reaches its weekly peak at 20:00 and the first hours of Saturday have a much higher intensity than other days. Sunday also begins with a high intensity of crashes, nearly as much as Saturday at 4:00, and then its intensity increases until 20:00, when it reaches its daily peak.

and road accidents and on Tuesdays at 3:00, the city is going through its coldest moment of the week. The intensity of crime increases by a magnitude of 7.5 times from its lowest to highest intensity, and 12.3 times in terms of crashes. This means that on a Friday night there will be a very high demand for security and emergency attention resources. At a city level, crime and crashes are somehow synchronised. We observe this by looking at just the time (and not the location) of where crime and crashes happen.

## The heartbeats of a tile

The tiles of a city have a different heartbeat with respect to crime and crashes (Fig 5). Although they follow a similar pattern to the city-level heartbeats, there are some tiles which reach their weekly crime peak on Saturdays, some which are more a morning tile, a plateau from morning to evening, or some which are night-intense tiles. Cities are not fractal in this regard: a tile is not a mini-city with a scaled version of the same crime or crashes heartbeat.

In terms of the heartbeat of crashes, we observe that some tiles tend to have a low intensity during the night, or their valley is not as pronounced, for other tiles. Furthermore, although the city overall has a high intensity of criminal activity during the nights (particularly Friday), this is the result of some -not all- tiles having a high intensity. Locally, crashes have a different heartbeat. Although on Sundays, for example, most tiles have a low intensity, there are tiles which have a low intensity during the week but reach their weekly peaks on a Sunday evening. Perhaps these tiles have a highway connecting to other cities, and Sunday evening activity corresponds to people returning from a weekend getaway to nearby cities and towns.

Nearby tiles frequently have highly correlated heartbeats, whereas distant tiles usually have less similar timely patterns, both in terms of crime and crashes. Take, for example, the tiles centred around Metro Juanacatlán (tile 1) and around Metro Velódromo (tile 2), located at a distance of more than 8 kilometres (Fig 6). In terms of crime, when tile 1 has a high intensity of crime on Tuesday, for example, tile 2 remains under its mean, but on Saturday (and Sunday), tile 2 has a peak of nearly 5 (and 2) standard deviations above its mean and tile 1 is way below. The crime heartbeat observed at a city level is the result of an aggregation process and the same steady pattern is not observed at a more local level. Local conditions matter. See the S1 File 1.5 for a more detailed analysis of the impact of physical distance.

## How similar are the heartbeats of different tiles?

In general, all tiles have a higher intensity of crime and crashes during the day and lower during the night, but not all tiles have the same (scaled, perhaps) heartbeat. For instance, some tiles peak during the mornings, some during the evening, or during the weekend, or do not peak at all, but have a roughly similar intensity during the day. We can use the heartbeats to measure how different is one tile to others.

Instead of an individual analysis of the many tiles, we group them based on the similarity of their heartbeats. For each tile, the crime heartbeat and the crashes heartbeat per area are considered and the tiles are clustered using $k$-means, a technique which has previously been used to cluster crashes based on its attributes [39]. The optimal number of clusters is obtained using

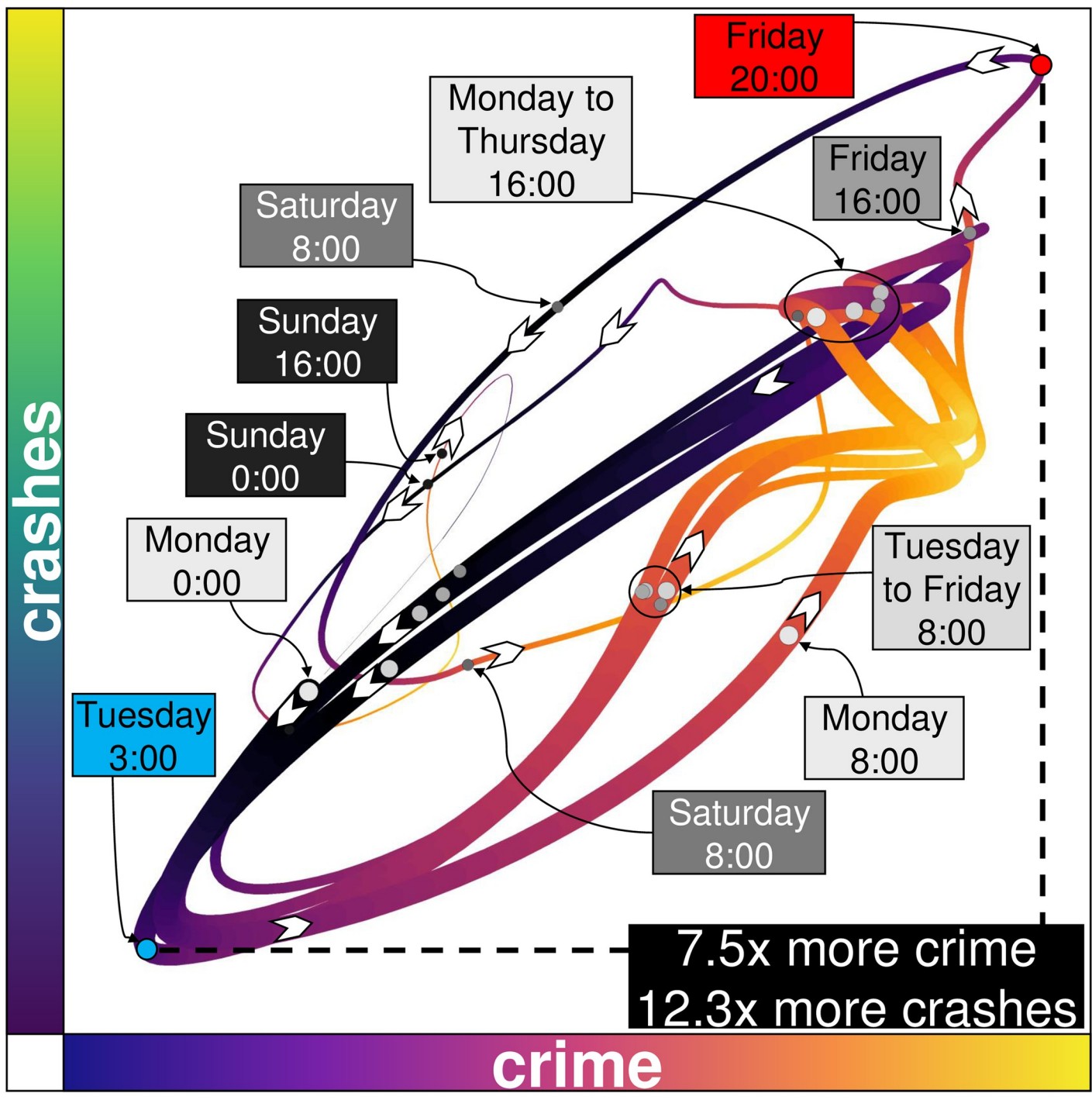

**Fig 4. Heartbeats associated with crime and crashes.** The horizontal axis shows the city crime heartbeat as the week evolves, and the vertical axis, the crashes heartbeat. The line is thick when the week begins on Monday at 0:00 and gets thinner as the 168 hours of the week progress. The shape of the heartbeats from Tuesday to Thursday is roughly the same, but Monday mornings are more intense in terms of crime. Both events reach their peak and valley simultaneously (the upper-right and lower-left corners).

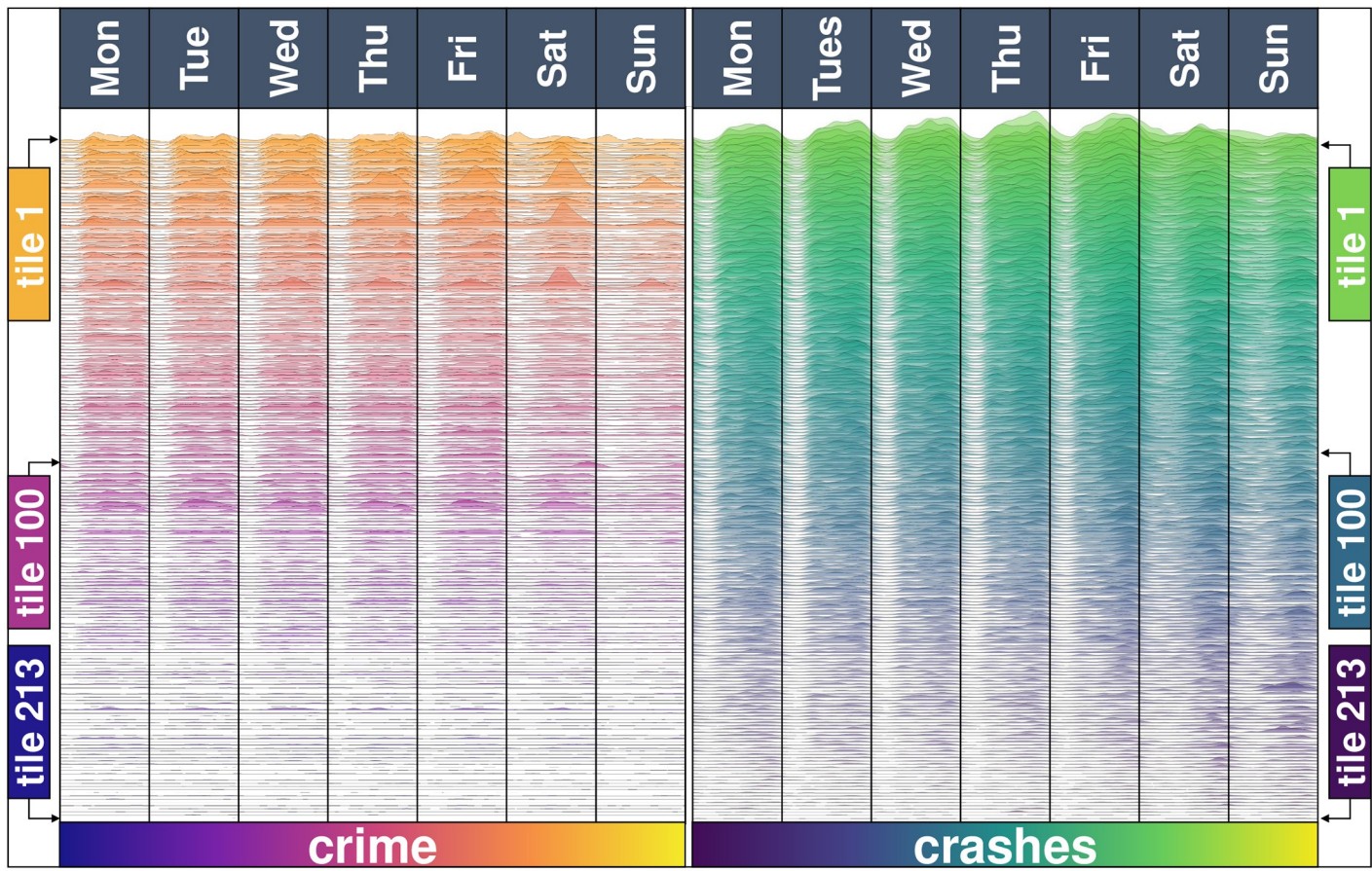

**Fig 5. The heartbeat of crimes and crashes for the 213 tiles in Mexico City.** The heartbeat of crimes is shown in the left panel, with the corresponding heartbeat of crashes on the right. Tiles are ordered from top to bottom by the intensity of crashes, which is also partially reflected in the ordering of the crime heartbeats.

the silhouette method (which is a measure of how similar is a heartbeat to its own cluster compared to others) [102, 103] via the gap statistic [104]. In summary, the gap statistic compares the variation within each cluster -that is, how distinct are the heartbeats of one cluster- and compares it against a random grouping of the data. Then, the optimal number of groups into which the tiles are divided maximise the distance between randomness and the actual clustering, so that the heartbeats are clustered as far away from the random uniform distribution [104]. The method gives $k^\star = 7$ groups for crime heartbeats and for crashes heartbeats.

The heartbeat of crime has a distinct shape around its mean (i.e., around the average intensity of the tiles). For example, groups A, B and C (see Fig 7 for the crime heartbeats and the corresponding cluster) are mostly located on the periphery of the city and have a small variation around the mean, particularly on Saturday and Sunday (especially Group C, located only on the edges of the city, with little variation on its crime heartbeat during the week). Groups D and E have a crime peak at around 20:00 during the weekdays, particularly on Friday and both groups are mostly located on central parts of the city, nearby the city centre. The main difference between groups D and E is a peak on Saturday afternoon (Group D) and a flat low intensity (Group E).

Group G is made up of a single tile centred around Metro Lagunilla. This tile includes Tepito, known for its traditional open-air market, but also for its counterfeiting of goods and robberies. On Saturdays, the market is very active, and the crime heartbeat reflects this. The

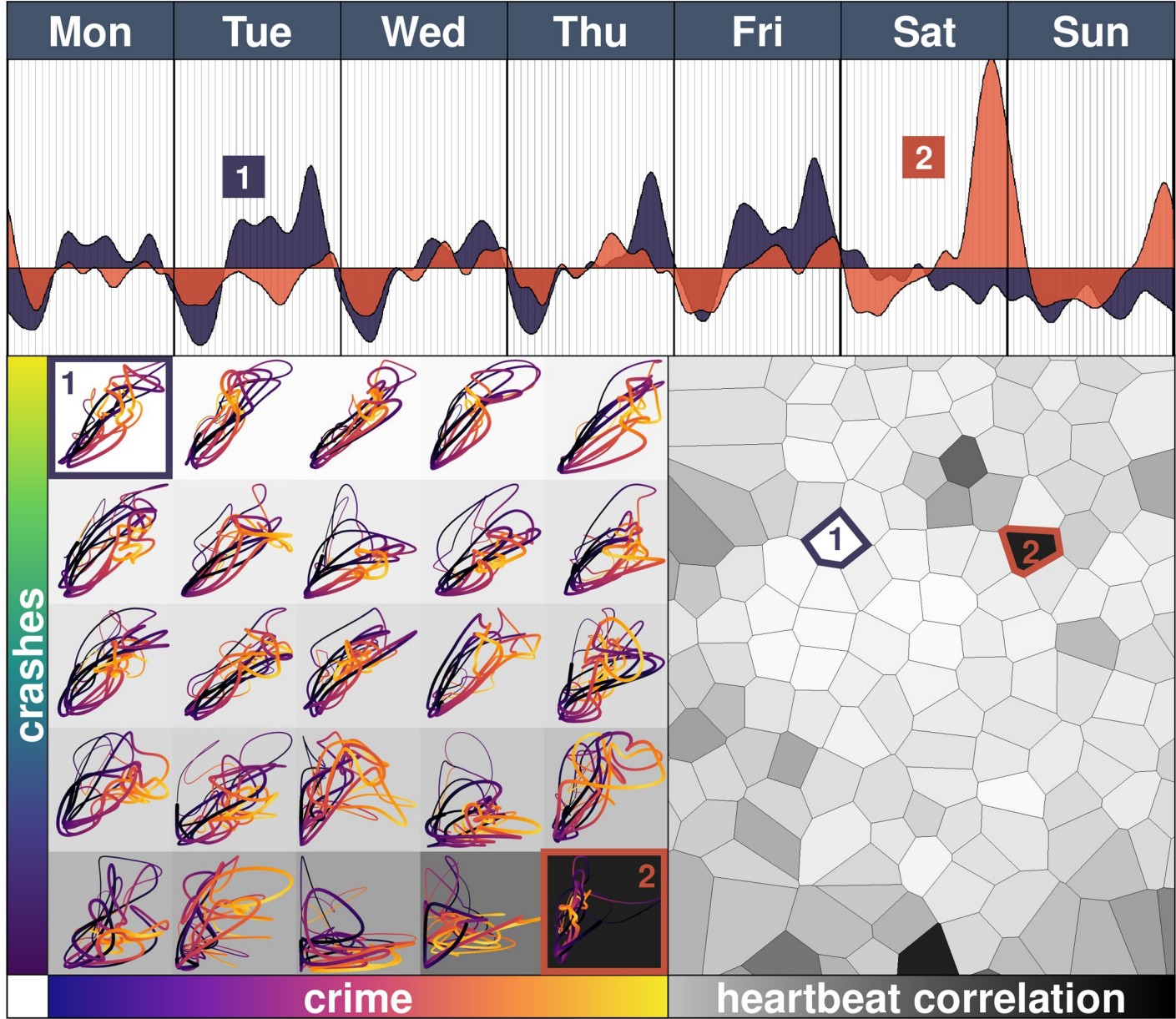

**Fig 6. Correlations of the crime heartbeats.** The top panel shows the crime heartbeat of two selected tiles, centred in Metro Juanacatlán (tile 1) and Metro Velódromo (tile 2) with a correlation of 0.1152 (one of the smallest pairwise correlations of the city). The left part shows 25 heartbeats of crime (horizontal axis) and crashes (vertical axis) as represented in Fig 4. The shading represents the correlation between the tile heartbeats (a lighter colour means a higher correlation) which is then represented in the right part of the figure, with a mark in tile 1 and tile 2.

market closes quite early, and so after 18:00, the flow of people (and therefore, its crime) is reduced considerably, which is also captured in the heartbeat.

Group F is made of two tiles, centred in Metro Hidalgo and Metro Isabel La Católica (next to Group G). These are highly active in tourism, and have a similar crime heartbeat from Monday to Friday as Group G, although incidents in this group also occur at later hours.

In terms of crime, more central groups (F and G) go below the mean at early hours, then the less central parts of the city go below the mean (D and E) and finally, the more peripheral tiles go below the mean much later, almost midnight (A, B and C). Thus, the central tiles go

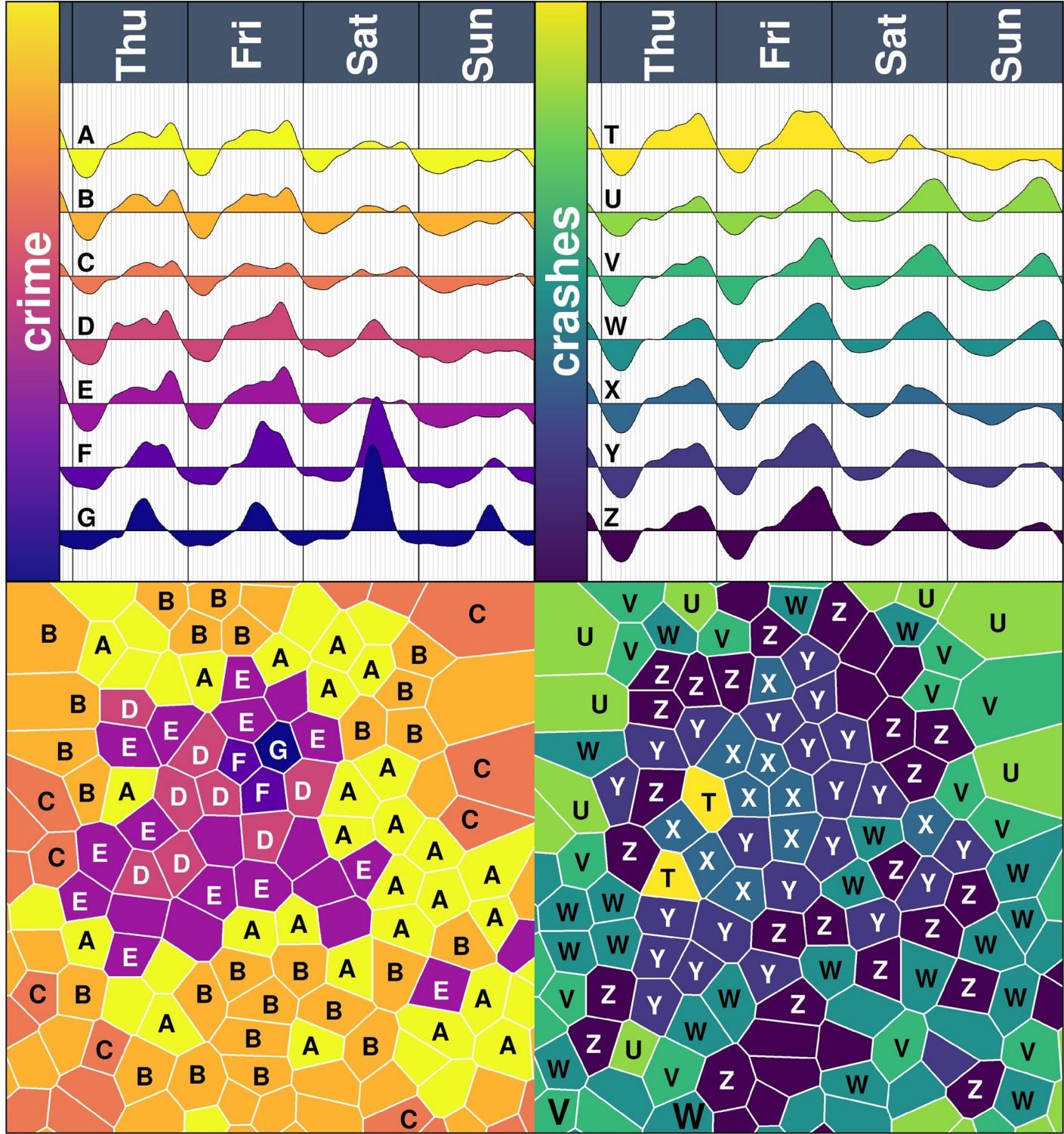

**Fig 7. Average heartbeat of crimes and crashes per cluster.** The top panel shows the average crime heartbeat (left) and the crashes heartbeat (right) of the seven different clusters from Thursday to Sunday (since Monday to Wednesday is similar to Thursday and so significant differences are observed within those four days). The bottom part shows the corresponding clustering map, where colours (and letters A-G for crime and T-Z for crashes) represent the clusters on top.

below the mean nearly three hours before the less central tiles and nearly six hours before the peripheral tiles. Similarly, in terms of crashes, the peripheral tiles (U, V and W) are mostly below the mean before 12:00, whereas central tiles (T, X, Y and Z) are above their mean from around 8:00.

In terms of crashes, the Western part of the city (the bottom-left part of the map of Fig 7) rather than marking areas, perhaps it identifies distinct avenues, including Periférico, Circuito Interior and Insurgentes, three main North-South corridors. Also, we see that the peripheral parts of the city are mostly on Group W, and the crashes heartbeats of the group are particularly pronounced on Saturday and Sunday evenings, which remains with a considerable high intensity on the first hours of Saturday and Sunday. Group Z is also on the more peripheral parts of the city, where the crashes heartbeat also peak considerably on Saturday and Sunday. Thus, on the peripheral areas of the city, crashes seem to have a higher intensity during the weekends. Group U has the highest variation on their Friday heartbeat, whilst Group V remains under the mean during the whole Sunday. Groups V and X have a high flow of pedestrians, as it includes parts of the core centre of the city, whilst Group Y marks some of the avenues with the highest car flow, such as Circuito Interior and Periférico.

The clustering obtained from comparing the heartbeat of crime and crashes across tiles are, to an extent, similar. In other words, two tiles which are in the same crime cluster are also likely to be in the same crash cluster. Formally, the Adjusted Rand Index is a measure of similarity between two clusterings. This is defined as the ratio of the number of times that two tiles are either in the same or in different clusters (meaning that the ratio discounts the number of times that there is a disagreement between the two clusterings) corrected for the randomness. We obtain an Adjusted Rand Index of 0.4416 and compare it to a null model where we randomly assign tiles to clusters. This yields a 95% interval between (0, 0.0162), and so we can conclude the two partitions are similar from a statistical perspective.

The heartbeats of crime and crashes are substantially different for a specific tile or group (see, for instance, the crime heartbeats of Group C and the crashes heartbeats of Group U, the peripheral tiles). Hence, it is not a simple case of "more people means more crime or more crashes" as they are not simultaneously peak or valley. Rather, crime and crashes are both complex phenomena and the spatio-temporal patterns we observe are the result of individual interactions and the local conditions.

## Why are there distinct heartbeats?

We have observed that there are distinct heartbeats in terms of crime and crashes, which can be compared using the correlation between tiles, which in general decreases as the physical distance increases (see the S1 File for the details 1.5). Some tiles have a higher intensity in the mornings, in the evenings or the weekends. This begs the question of why some tiles have peaks in crime or crashes at different times. What, if anything, do the group of tiles with weekend peaks have in common? Here, we use the amenity mix of the tiles [105] to see if, for example, a tile which has many schools has morning peaks.

Data from the Mexican Economic Census (collected by the National Institute of Statistics and Geography, updated in April 2020) gives the location of nearly half a million distinct economic units in Mexico City (for example pharmacies, schools, factories, bars, restaurants and offices). It also includes some of informal businesses (such as street food restaurants, if they have any type of infrastructure in the street). The dataset also includes an approximation of the number of workers in each economic unit. We classified the units into six main categories: Services, Education, Leisure, Offices, Manufacture and Health (see the S1 File for the details 1.8).

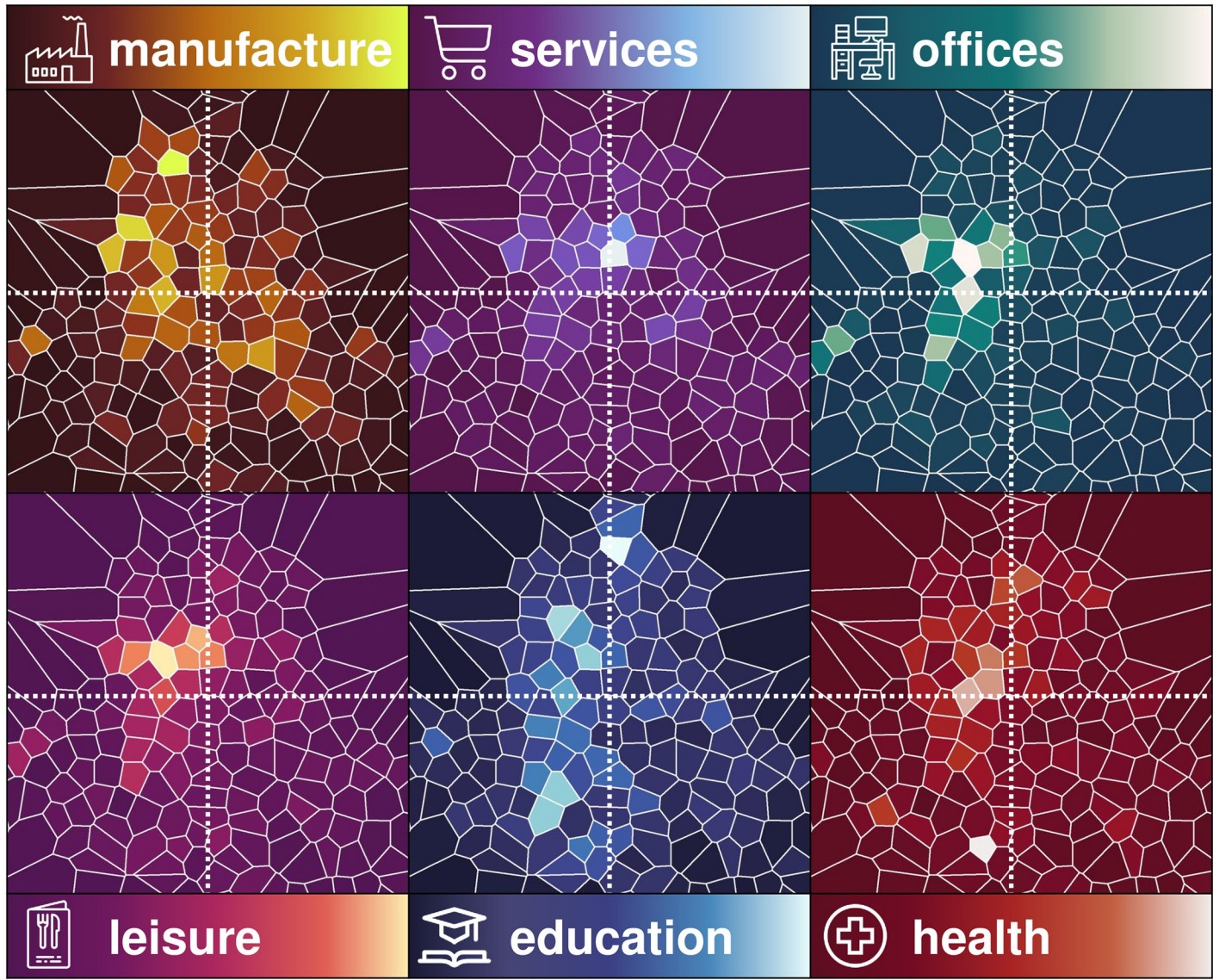

**Fig 8. Intensity of distinct types of amenities per area.** The maps show the intensity of distinct types of amenities per unit area. We observe that some parts of the city have a high intensity of offices (including the city centre and the financial district on the extreme West), some parts with many leisure spaces (such as bars and restaurants in Condesa-Roma-Zona Rosa), there are some school neighbourhoods (including universities such as IPN in the North part and UNAM across many tiles in the South part) and the hospital neighbourhood in the South, whilst many factories are in the North of the city.

The size (workforce) of each unit is used as a proxy for the number of people who attend that unit. For each tile, we use the number of users per unit area of distinct types of economic units and compare it against the tile with the highest number of users, so that we deploy a relative scale (a tile has two times more education units than another tile, for example).

Tiles have many economic units which belong to different amenity types. This enables us to see (per unit area) if a tile has many offices, many schools, many leisure units and so on (Fig 8). Notice that it is not possible to compare the number of "users" between distinct types of amenities, since for offices or for manufacturing, for example, its size is a fairly good representation of the number of employees, but for restaurants, for example, we do not know the number of customers (but we know the number of employees) and for schools, we know the staff

size but not the student size. However, we can distinguish if a tile has many school staff (and therefore students) as compared to other tiles, and we can identify tiles with many leisure units or health units and so on.

As we observed before in the tiles of Metro Juanacatlán and Metro Velódromo (Fig 6) they have substantially distinct crime heartbeats and they also have substantially different amenity mix. The tile Metro Juanacatlán has nearly 6 times more leisure spaces and 5 work places per area than the tile Metro Velódromo, as well as 3.3 more services and 3 times more education size. The tile Metro Juanacatlán includes parts of a leisure neighbourhood (Roma-Condesa) and the tile Metro Velódromo covers three large stadiums (Foro Sol, Palacio de los Deportes and Velódromo Olímpico) which are frequently used as venues for large events and concerts. Thus, it is not surprising that the Metro Velódromo heartbeat has peaks during Saturday and Sunday nights, as opposed to the Metro Juanacatlán heartbeat, which has peaks from Monday to Friday evenings.

### A different district, a different heartbeat

For the seven crime clusters, a distinct amenity mix is observed (Fig 9). Groups A and B have few offices and service units, but many education and health units. This means that they are mostly residential tiles, with a very similar heartbeat from Monday to Friday exhibiting two peaks in the morning and evening, and low intensity on weekends. Group C has few amenities, and is formed mostly of peripheral tiles of the city. Groups with a large number of leisure units, meaning restaurants, bars and cafes (groups D, F and G) have an increasing peak from Monday to Friday (meaning, their peak is higher on Tuesday than it is on Monday, and so on) and also a peak on Saturdays. The difference between group D and E is that group E has more manufacturing units, but group D is more mixed, with offices and other amenities.

In terms of crashes, we see that groups which have a high density of office and education units (T, X and Y) tend to have a peak Monday to Friday around lunchtime. Also, the clusters which have few leisure units, particularly Group U and V, do not have such a marked peak on Friday night as the other groups. Groups with few amenities in all categories (mostly residential areas, e.g., U, V and W) have an intense peak on Saturday and Sunday after 12:00, whereas the other groups exhibit a low intensity of crashes on Sunday.

Previously we clustered tiles based on the temporal pattern (heartbeat) of crimes and crashes. Now we take a different approach, and cluster tiles based on their amenity mix. This results in seven clusters of tiles, each corresponding to a distinct amenity 'signature'. We can compare the partition of tiles obtained from the amenity-based clustering to that obtained previously from the crime-heartbeat-based clustering, and obtain an Adjusted Rand Index of 0.3912. Similarly, we can compare the amenity-based partition to the crash-heartbeat-based partition, which yields an Adjusted Rand Index of 0.3492. Both indices are above the rejection interval, implying that the spatial pattern associated with temporal patterns of crimes and crashes is similar to that of amenities.

With distinct values of $\tau$, results also show that clustering tiles based on their crime heartbeat and their crashes heartbeat, as well as with their amenity mix yield similar results (see the S1 File for a more detailed analysis by changing the size of $\tau$ and the number of tiles considered 1.6).

### How similar are different times of the week?

We have used the heartbeat of crimes and crashes to compare, correlate and cluster tiles. These heartbeats also enable us to analyse different times of the week. E.g., how similar is the spatial distribution of crimes on a Monday morning compared to a Thursday evening? Or,

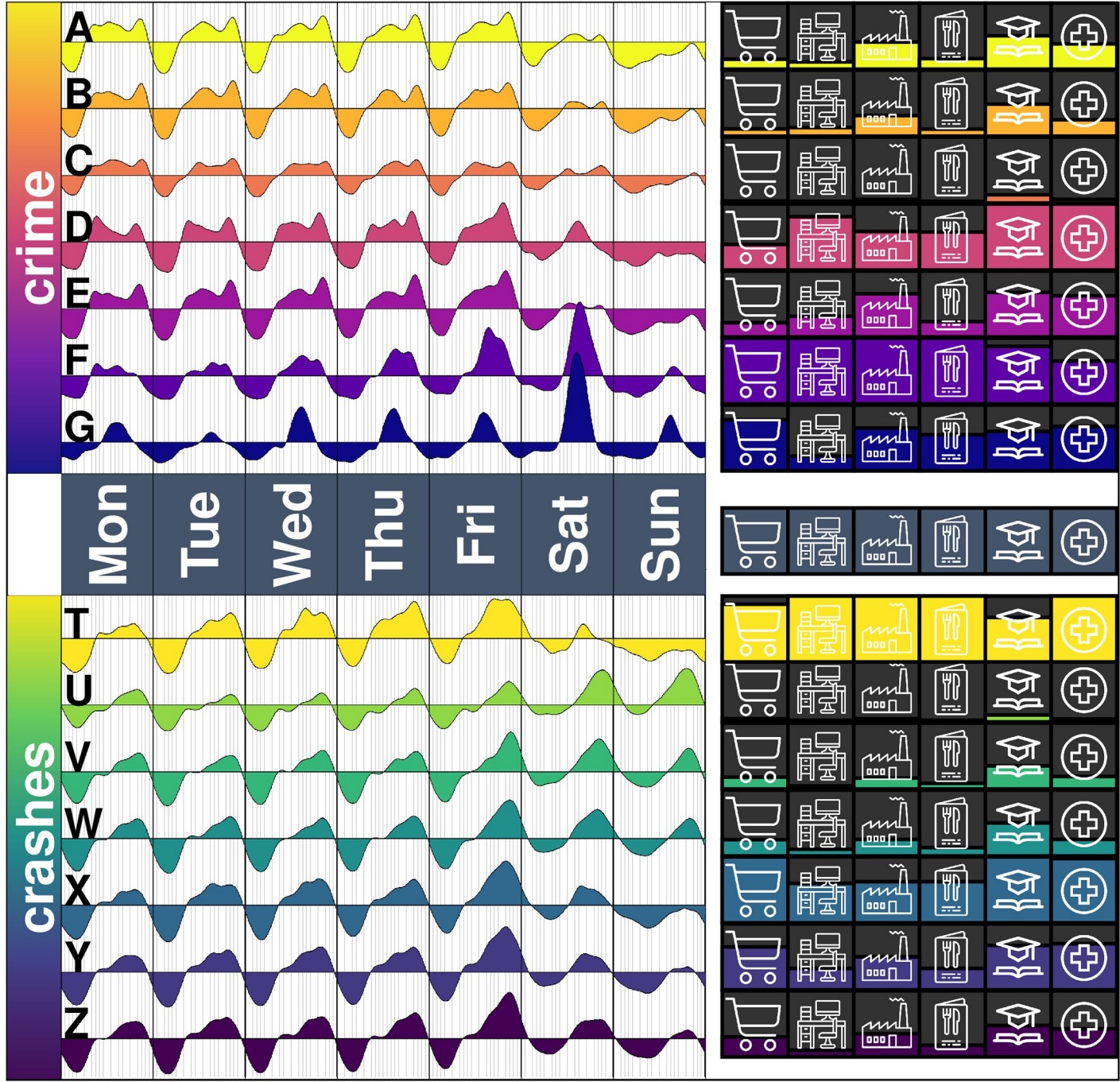

**Fig 9. Amenities and heartbeats per cluster.** The left panel is the heartbeat of crime (top) and the heartbeat of crashes (bottom) for the seven clusters. The right panel is the amenity mix of each cluster.

equivalently, if a tile has a valley (or a peak) on a Monday morning, can we say anything about a Thursday evening? Considering times of the week as the unit of observation (for this exercise, we divided the 168-hour cycle of the week into 1,024 units of nearly 10 minutes each) and the heartbeat of each tile (intensity of crimes or crashes) as the data for that observation, we compute the correlation between the units (Fig 10).

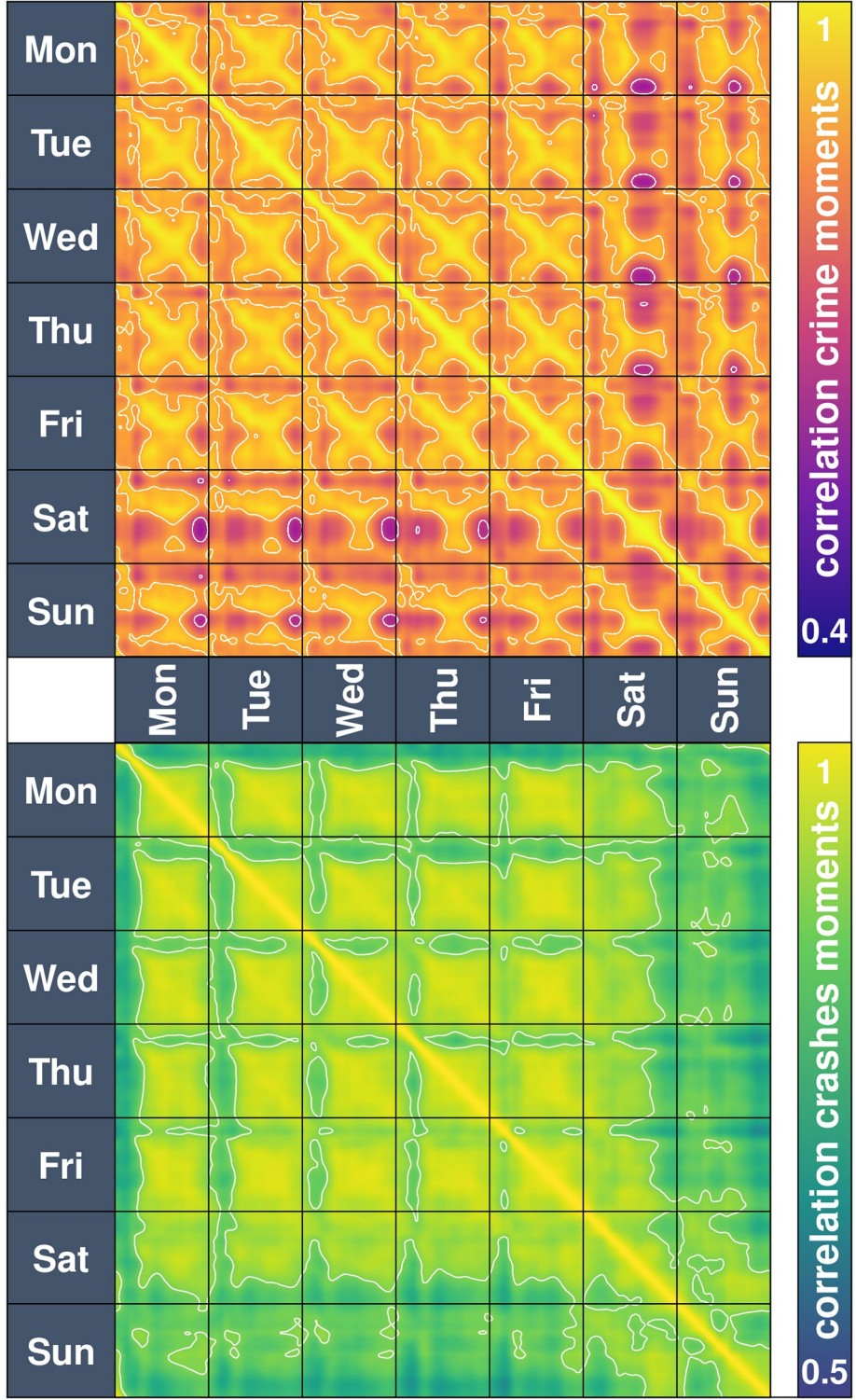

**Fig 10. Correlations of distinct moments of the week.** The top panel shows the correlation of different crime moments, which range between values of 0.4 for a small correlation (between a Monday night and a Saturday evening, for instance) and 1 for a high correlation. The bottom panel shows the correlation of different crashes moments, which range between values of 0.5 for a small correlation (for example, early hours of a Wednesday and Sunday evening) and 1 for a high correlation. There is a high correlation between every moment and its surrounding tile. The high values around the diagonal are the result of the Kernel density estimation of the heartbeats.

Results show that in terms of the distribution of crime, there is a clear repetitive pattern. Indeed, Monday mornings are similar to Tuesday mornings, Wednesday mornings and so on. From Monday to Friday, all days are similar in terms of where in the city there are high and low-intensity areas. Early Saturdays look like early Sundays, but late Saturdays are more correlated to late Fridays. Interestingly, the days of the week (particularly Monday to Friday) are similar at the exact same hour (so, Monday at 11:00, say, has a similar crime distribution as Wednesday at 11:00). But we also observe an opposite pattern in that early Wednesdays are similar to late Thursdays or late Mondays (the crosses in Fig 10 correspond to a correlation in the direction of time but in the opposite direction as well). The similarity follows the direction of time (so that a Monday at 9:00 is similar to Tuesday at 9:00; Monday at 17:00 is similar to Tuesday at 17:00) but also in the opposite direction of time (with a delay), and both directions eventually converge at around 14:30. This means that Monday at 18:00 is similar to Tuesday at 18:00 but also to Monday at 18:00 is similar to Monday at 11:00, and Monday at 19:00 is similar to Monday at 10:00. This is likely due to commutes corresponding to similar locations (but different direction) on those opposite times. Likely, this is the result of people commuting back from their daily activities, and so opportunities for committing a crime are similar during the mornings and the evenings. In other words, the rush hour in the mornings is likely similar to the rush hour in the evenings (although the flow goes in the opposite direction), but this hypothesis requires further analysis.

In terms of crashes, there is a high correlation in their distribution, particularly from Monday to Friday from 8:00 to 23:00 (or even to 1:00 the day after). Sundays are highly correlated to Saturdays at roughly the same hour (meaning that early Saturdays and early Sundays have a similar distribution of the crashes). In the case of crashes, blocks of high correlation are observed and the crosses (observed in crime) are much less defined.

Notice that in terms of crime, the early hours of any day have less correlation with other days. This effect is even more pronounced for crashes. For instance, the early hours of Monday or Tuesday are minimally correlated with other times of the week. Crime and crashes are much less frequent during the early hours of the day (Fig 1), and have a distinct spatial distribution relative to the rest of the week. See the S1 File 1.7 for a similar analysis using different spatial units.

## Conclusions

Underlying social patterns are complex, difficult to capture, model and quantify. Here we propose a methodology which enables us to capture daily and weekly cycles in point events such as crime and road accidents. The temporal aspect of such events is frequently analysed using time series, typically using weekly averages as the unit of observation. We focus on analysing the 168-hour cycle of the week, a more natural time-span for the analysis of the temporal patterns of crime and crashes in cities.

We refer to the temporal intensity of crime and crashes over a week as a heartbeat. By computing the heartbeats of crime and crashes in Mexico City, we show that both events reach their weekly peak and valley simultaneously. This is not a result of all tiles having the same heartbeat, but there exists a spatio-temporal pattern with respect to both types of events.

Our results have implications for the security and emergency services of a city. Whilst both events are frequently analysed separately, we have shown that they are largely synchronised. Therefore, it is likely that when the police have the highest demand due to crime, response times might be increased due to crash demand, and vice-versa. Throughout the week, more or less police are needed for an optimal response time to 911 calls in the case of a crime or a

crash. Also, their location should correspond to the local intensity which is related to the distance to the city centre and the amenity mix of each part of the city.

We show that the heartbeat of crime and the heartbeat of crashes differs across the city. We measure the similarity of tiles and observe that nearby tiles tend to have more similar heartbeats and this similarity decreases as distance increases. Thus, we conjecture that there is some underlying urban pattern which gives rise to this heartbeat similarity. For example, tiles in peripheral parts of the city might be synchronised due to commuting patterns, whereas tiles in central parts might avoid rush hour and therefore have less crime or crashes late at night. By clustering tiles, we are able to confirm our theory and notice that the intensity of crime tends to be higher late at night in more peripheral parts, while the intensity of crashes tends to increase during the weekend. By looking at the amenity mix of distinct parts of the city, we find that there are some tiles which are more residential, commercial, or with more leisure activities (bars and restaurants) or working spaces. We find that the heartbeat of crime in parts with more leisure activities tends to have a higher intensity on Friday and Saturday and that areas with a lower amenity mix (more residential areas) have a higher intensity of crashes during weekends. Again, grouping tiles based on their crime heartbeat (or the crashes heartbeat) and on the amenity mix gives a partition which is similar, statistically speaking.

Finally, the heartbeats enable us to analyse temporal patterns. We find that the distribution of crime on a Monday morning is similar to the distribution on a Tuesday morning, and Wednesday, but surprisingly, it is also similar to the distribution of crime on a Monday evening. The high similarity follows the direction of time but also in the opposite direction (with a delay). This is likely due to return commutes corresponding to similar locations (but opposite direction) on those times. In terms of crashes, instead of two opposite directions, we find blocks, which begin at around 7:00 and finish at around 2:00 the day after. Crashes from 2:00 to 7:00 are rare and happen at patterns which are not as repetitive as the rest of the day, as correlations are at the lowest.

Here we only considered robberies as we have more certainty about their time and location. Likely, the heartbeats of different types of crime are also different and give us more information with respect to the spatial distribution of crime, for instance, gang fights. The same is true in the case of road accidents, where the heartbeat observed when pedestrians are involved is different to when cyclists or only vehicles. Also, further work is required to detect whether heartbeats exhibit some seasonality (for instance, during holidays), some stability (comparing, for example, between distinct years) and shocks (for instance, due to the pandemic).

Cities have a heartbeat which emerges as millions of citizens synchronise their daily routines. Here we observed the heartbeat of crime and the heartbeat of crashes, but other activities should have a heartbeat as well, such as transactions and purchases, messages, calls, and posts on social media or even streaming viewers or $CO_2$ emissions. Although the heartbeats observed in Mexico City in terms of crime and road accidents are not necessarily the same in other cities, our weekly schedules govern patterns observed for all social activities.

## Supporting information

**S1 File.**
(PDF)

## Acknowledgments

We thank Jerónimo Mohar Volkow for his valuable comments.

## Author Contributions

**Conceptualization:** Rafael Prieto Curiel.

**Data curation:** Rafael Prieto Curiel.

**Formal analysis:** Rafael Prieto Curiel, Jorge Eduardo Patino, Juan Carlos Duque.

**Funding acquisition:** Neave O'Clery.

**Investigation:** Rafael Prieto Curiel.

**Methodology:** Rafael Prieto Curiel, Jorge Eduardo Patino, Juan Carlos Duque.

**Supervision:** Juan Carlos Duque, Neave O'Clery.

**Visualization:** Rafael Prieto Curiel, Juan Carlos Duque.

**Writing – original draft:** Rafael Prieto Curiel, Jorge Eduardo Patino, Juan Carlos Duque.

**Writing – review & editing:** Rafael Prieto Curiel, Jorge Eduardo Patino, Juan Carlos Duque, Neave O'Clery.

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
