## [Decision Letter · Decision Letter 0]

17 Nov 2020

PONE-D-20-27934

The heartbeat of the city

PLOS ONE

Dear Dr. Prieto Curiel,

Thank you for submitting your manuscript to PLOS ONE. After careful consideration, we feel that it has merit but does not fully meet PLOS ONE’s publication criteria as it currently stands. Therefore, we invite you to submit a revised version of the manuscript that addresses the points raised during the review process.

Please try to revise your manuscript and respond to all the reviewers' comments.

We look forward to receiving your revised manuscript.

Kind regards,

Quan Yuan, Ph.D.

Academic Editor

PLOS ONE

Journal Requirements:

Reviewers' comments:

Reviewer's Responses to Questions

**Comments to the Author**

1. Is the manuscript technically sound, and do the data support the conclusions?

Reviewer #1: Yes

Reviewer #2: Yes

2. Has the statistical analysis been performed appropriately and rigorously? 

Reviewer #1: Yes

Reviewer #2: Yes

3. Have the authors made all data underlying the findings in their manuscript fully available?

Reviewer #1: Yes

Reviewer #2: Yes

4. Is the manuscript presented in an intelligible fashion and written in standard English?

Reviewer #1: Yes

Reviewer #2: Yes

5. Review Comments to the Author

Reviewer #1: There are several questions that the author should explain or add:

1. Are there any distinction between the tiles in the division of traffic districts?

2. Is the number of crimes and traffic accidents representative?

3. What is the correlation and difference between traffic based crime and traffic accident?

4. In Figure 10, what is the basis for the correlation coefficient to take 0.4 and 0.5? What's the point?

5. For the statistical results, the specific figures shall be reflected and the charts shall be combined.

Reviewer #2: The authors are recommended to conduct a more comprehensive literature review work. The introduction section did not clearly demontrate the unique contributions of this study over the existing studies. Moreover, the clear defination of the problem of this study is missing.

6. PLOS authors have the option to publish the peer review history of their article (what does this mean?). If published, this will include your full peer review and any attached files.

Reviewer #1: **Yes: **Xuejian Kang

Reviewer #2: No

---

## [Author Response · Author response to Decision Letter 0]

4 Dec 2020

A full point-by-point answer to the reviewers is included in the submission.

---

## [Decision Letter · Decision Letter 1]

26 Jan 2021

The heartbeat of the city

PONE-D-20-27934R1

Dear Dr. Prieto Curiel,

We’re pleased to inform you that your manuscript has been judged scientifically suitable for publication and will be formally accepted for publication once it meets all outstanding technical requirements.

Kind regards,

Quan Yuan, Ph.D.

Academic Editor

PLOS ONE

Additional Editor Comments (optional):

Reviewers' comments:

Reviewer's Responses to Questions

**Comments to the Author**

1. If the authors have adequately addressed your comments raised in a previous round of review and you feel that this manuscript is now acceptable for publication, you may indicate that here to bypass the “Comments to the Author” section, enter your conflict of interest statement in the “Confidential to Editor” section, and submit your "Accept" recommendation.

Reviewer #1: All comments have been addressed

Reviewer #2: All comments have been addressed

2. Is the manuscript technically sound, and do the data support the conclusions?

Reviewer #1: (No Response)

Reviewer #2: Yes

3. Has the statistical analysis been performed appropriately and rigorously? 

Reviewer #1: (No Response)

Reviewer #2: Yes

4. Have the authors made all data underlying the findings in their manuscript fully available?

Reviewer #1: (No Response)

Reviewer #2: No

5. Is the manuscript presented in an intelligible fashion and written in standard English?

Reviewer #1: (No Response)

Reviewer #2: Yes

6. Review Comments to the Author

Reviewer #1: (No Response)

Reviewer #2: The authors have carefully addressed all the comments and issues that have been raised by the reviewers.

7. PLOS authors have the option to publish the peer review history of their article (what does this mean?). If published, this will include your full peer review and any attached files.

Reviewer #1: No

Reviewer #2: No

---

## [Editor Report · Acceptance letter]

28 Jan 2021

PONE-D-20-27934R1 

The heartbeat of the city 

Dear Dr. Prieto Curiel:

I'm pleased to inform you that your manuscript has been deemed suitable for publication in PLOS ONE. Congratulations! Your manuscript is now with our production department. 

Kind regards, 

on behalf of

Dr. Quan Yuan 

Academic Editor

PLOS ONE